# Effect of belt electrode-skeletal muscle electrical stimulation on immobilization-induced muscle fibrosis

Yuichiro Honda[1], Natsumi Tanaka[2,3], Yasuhiro Kajiwara[2,3], Yasutaka Kondo[4], Hideki Kataoka[3,5], Junya Sakamoto[1], Ryuji Akimoto[6], Atsushi Nawata[7], Minoru Okita[1,3]*

1 Institute of Biomedical Sciences (Health Sciences), Nagasaki University, Nagasaki, Japan, 2 Department of Rehabilitation, Nagasaki University Hospital, Nagasaki, Japan, 3 Department of Physical Therapy Science, Nagasaki University Graduate School of Biomedical Sciences, Nagasaki, Japan, 4 Department of Rehabilitation, Japanese Red Cross Nagasaki Genbaku Hospital, Nagasaki, Japan, 5 Department of Rehabilitation, Nagasaki Memorial Hospital, Nagasaki, Japan, 6 Research and Development Division, HOMER ION Co., Ltd., Shibuya, Tokyo, Japan, 7 Medical Engineering Research Laboratory, ALCARE Co., Ltd., Sumida, Tokyo, Japan

* mokita@nagasaki-u.ac.jp

**Data Availability Statement:** All relevant data are within the manuscript and its Supporting Information files.

## Abstract

### Purpose

Macrophage accumulation in response to decreasing myonuclei may be the major mechanism underlying immobilization-induced muscle fibrosis in muscle contracture, an intervention strategy suppressing these lesions is necessary. Therefore, this research investigated the effect of belt electrode-skeletal muscle electrical stimulation (B-SES), a new electrical stimulation device, to the macrophage accumulation via myonuclei decrease in immobilization-induced muscle fibrosis.

### Materials and methods

18 Wistar male rats were divided into the control group, immobilization group (with plaster cast fixation to immobilize the soleus muscles in a shortened position for 2 weeks), and B-SES group (with muscle contractile exercise through B-SES during the immobilization period). B-SES stimulation was performed at a frequency of 50 Hz and an intensity of 4.7 mA, muscle contractile exercise by B-SES was applied to the lower limb muscles for 20 minutes/session (twice a day) for 2 weeks (6 times/week). The bilateral soleus muscles were used for histological, immunohistochemical, biochemical, and molecular biological analyses.

### Results

The number of myonuclei was significantly higher in the B-SES group than in the immobilization group, and there was no significant difference between the B-SES and control groups. The cross-sectional area of type I and II myofibers in the immobilization and B-SES groups was significantly lower than that in the control group, and the cross-sectional area of type I myofibers in the B-SES group was higher than that in the immobilization group. However,

**Funding:** This work was supported by Ministry of Education, Science, Sports, and Culture of Japan (https://www.mhlw.go.jp/stf/seisakunitsuite/bunya/hokabunya/kenkyujigyou/index.html) in the form of grants awarded to YH (16K16427, 19K19795), HOMER ION Co., Ltd. (http://www.homerion.co.jp) in the form of a salary for RA, and ALCARE Co., Ltd. (https://www.alcare.co.jp/) in the form of a salary for AN. The specific roles of these authors are articulated in the 'author contributions' section. The funders had no role in study design, data collection and analysis, decision to publish, or preparation of the manuscript.

Atrogin-1 and MuRF-1 mRNA expression in the immobilization and B-SES groups was significantly higher than those in the control group. Additionally, the number of macrophages, IL-1β, TGF-β1, and α-SMA mRNA expression, and hydroxyproline expression was significantly lower in the control and B-SES groups than those in the immobilization group.

## Conclusion

This research surmised that muscle contractile exercise through B-SES prevented immobilization-induced muscle fibrosis, and this alteration suppressed the development of muscle contracture.

## Introduction

Muscle contracture, caused by immobilization of joints through excessive rest, limits daily activities and interferes with rehabilitation [1]. Reduced muscle extensibility decreases joint mobility, thus contributing to muscle contracture. In our previous study, muscle fibrosis induced by collagen overexpression was the main cause of reduced muscle extensibility [2]. In addition, our laboratory also showed in another study that the accumulation of macrophages, which induces the differentiation of fibroblasts into myofibroblasts via interleukin (IL)-1β/transforming growth factor (TGF)-β1 signaling, might affect the incidence of immobilization-induced muscle fibrosis in muscle contracture [3]. The same study demonstrated that muscle atrophy and muscle fibrosis occurred simultaneously during early immobilization [3]. Previous research has shown that myonuclei are removed from the cytoplasm during muscle atrophy [4]. A study by Reilly and Franklin also suggested that the unnecessary cytoplasm generated by decreasing myonuclei was eliminated under muscle atrophy conditions [5]. Additionally, macrophages play a key role in the clearance of unnecessary muscle cytoplasm [6]. From these previous studies, the macrophage accumulation in response to decreasing myonuclei may be the major mechanism underlying muscle fibrosis. Therefore, our laboratory hypothesized that an intervention strategy suppressing these lesions is necessary to prevent immobilization-induced muscle fibrosis.

Electrical stimulation therapy has been utilized as a therapeutic intervention and a functional substitute for voluntary muscle contraction in patients [7]. The common patterns of muscle contraction by electrical stimulation are twitch and tetanic contractions, with a stimulus frequency of 1–10 Hz and 50–100 Hz in rat skeletal muscle, respectively [8, 9]. The importance of active contractions for the maintenance of skeletal muscle is well-documented [10], and electrical stimulation that generates tetanic contractions in the skeletal muscles of rats have been shown to prevent muscle atrophy [11]. Although a conventional electrical stimulation device often energizes skeletal muscle through a monopolar electrode, this method has several problems. In fact, as electrical stimulation using the monopolar electrode stimulates only superficial skeletal muscles, the efficiency of contraction of deep skeletal muscle through this method remains unclear. In some situations of electrical intervention, sufficient current (power) for muscle contractile exercise may not be obtained owing to the limited size of the electrodes, whereas excessive current can cause pain. In brief, an innovative electrical stimulation method that has a safety large current-carrying capacity is needed. Recently, belt electrode-skeletal muscle electrical stimulation (B-SES; HOMER ION, Tokyo, Japan) was developed as a novel method of electrical stimulation therapy. An advantage of B-SES is that the entire belt area is an electrode, B-SES can deliver electricity to the entire lower limb [12].

Additionally, it is less likely to cause pain during muscle contractile exercise owing to the dispersed distribution of electricity in this intervention. Thus, B-SES has high efficiency of supplying electrical stimulation, this rehabilitation intervention may be more effective in suppressing muscle atrophy than electrical stimulation achieved with a monopolar electrode [12].

This investigation hypothesized that B-SES may be effective not only for muscle atrophy but also for muscle fibrosis in immobilized skeletal muscle. Therefore, the present study analyzed 2-week immobilized rat soleus muscles to confirm the effect of muscle contractile exercise through B-SES on immobilization-induced muscle fibrosis.

## Materials and methods

### Animals

Eight-week-old male Wistar rats (CLEA Japan Inc., Tokyo, Japan) were maintained at the Center for Frontier Life Sciences at Nagasaki University. The rats were maintained in $30 \times 40 \times 20$-cm cages (2 rats/cage) and exposed to a 12-h light-dark cycle at an ambient temperature of 25°C. Food and water were available *ad libitum*. In this investigation, 18 rats (266.9 ± 12.4 g) were randomly divided into the experimental group (n = 12) and the control group (n = 6). In the control group, the rats were normally maintained without treatment and intervention. In the experimental group, the ankle joints of the animals were subjected to the immobilization process detailed in our previous studies [4]. Briefly, the animals in the experimental group were anesthetized with the combination of the following anesthetic agents: 0.375 mg/kg medetomidine (Kyoritu Pharma, Tokyo, Japan), 2.0 mg/kg midazolam (Sandoz Pharma Co., Ltd., Tokyo, Japan), and 2.5 mg/kg butorphanol (Meiji Seika Pharma, Tokyo, Japan). Then, both ankle joints of each rat were fixed in full plantar flexion with plaster casts to immobilize the soleus muscle in a shortened position for 2 weeks. The plaster cast, which was fitted from above the knee joint to the distal foot, was changed weekly because of loosening due to muscle atrophy. Additionally, the experimental groups were divided into the immobilization group (n = 6; with immobilized treatment only) and the B-SES group (n = 6; with immobilized treatment and muscle contractile exercise through B-SES). The experimental protocol was approved by the Ethics Review Committee for Animal Experimentation of Nagasaki University (approval no. 1404161137). All experimental procedures were performed under anesthesia, and all efforts were made to minimize suffering.

### Protocol for B-SES

Cyclic muscle tetanus contraction was performed using an electrical stimulator for small animals (Homer Ion). The electrical stimulator consists of a control unit (for setting the stimulus cycle, frequency, and intensity) and a belt electrode (Fig 1A). The rats in the B-SES group were anesthetized, and the electrical stimulator was connected to a belt electrode. The belt electrodes were wrapped around the proximal thigh and distal lower leg, and the bilateral lower-limb skeletal muscles were subjected to B-SES with cast removed (frequency, 50 Hz). However, it was necessary to determine the optimum stimulus intensity and time before starting the experiment. Therefore, this study conducted preliminary experiments to determine the stimulation intensity and time, as outlined below.

**Preliminary experiment for stimulus intensity determination.** Three rats were used in this preliminary experiment. The stimulus intensity was gradually increased, and the plantar flexor muscle strength in the middle position of the ankle joint was measured with a force gauge. Next, the 100% maximal voluntary contraction (MVC) was determined, and the 60% MVC (most effective in preventing muscle atrophy) was calculated. As a result, our

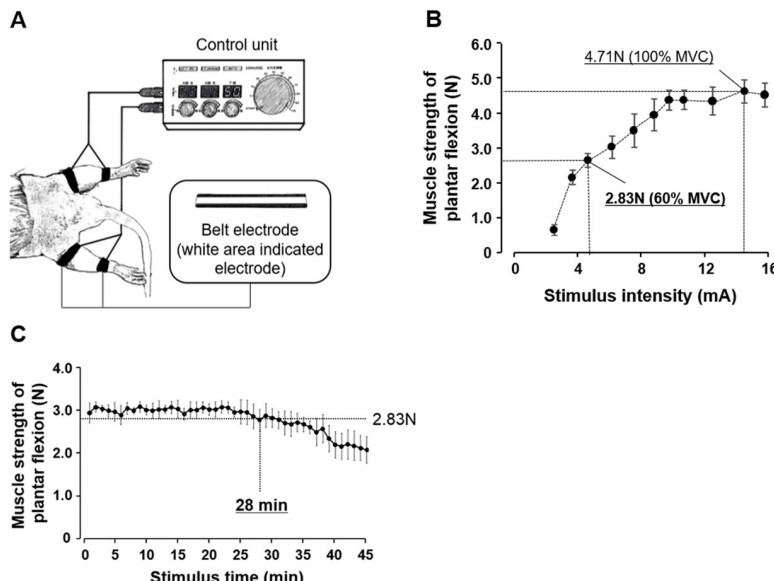

**Fig 1. Diagram of the electrical stimulator for small animals and results of preliminary experiments to determine the stimulation intensity and time.** (A) The control unit sets the duty cycle, frequency, and intensity. The white area of the belt is the electrode. The belt electrodes were wrapped around the proximal thigh and distal lower leg, and then the left and right soleus muscles were subjected to belt electrode-skeletal muscle electrical stimulation. (B) Preliminary experiment to determine the stimulus intensity. The 100% maximal voluntary contraction (MVC) was 4.7 N, whereas the 60% MVC was 2.8 N. The results indicated that 4.7 mA corresponded to 60% MVC. (C) Preliminary experiment to determine the stimulus time. In this experiment, the changes in muscle strength of plantar flexion were observed. The results indicated that the muscle strength of plantar flexion never decreased to < 2.83 N until 28 min after the electrical stimulation.

preliminary experiment confirmed that 4.7 mA corresponded to a 60% MVC (Fig 1B). There-fore, this research defined the stimulus intensity as 4.7 mA.

**Preliminary experiment for stimulus time determination.** Three rats were used in this preliminary experiment. The muscle strength of plantar flexion was measured (7–8 measure-ments per minute), and the stimulus time to reach < 2.83 N (60% MVC) was identified. As a result, our preliminary experiment indicated that the muscle strength of plantar flexion never decreased to < 2.83 N until 28 min after starting the electrical stimulation (Fig 1C). Therefore, this investigation decided the stimulation time to be 20 min (without muscle fatigue).

In the B-SES protocol of the present study, the stimulus frequency was 50 Hz, the stimulus intensity was 4.7 mA, the duty cycle was 2 s (do)/6 s (rest), and the stimulus time was 20 min. Electrical stimulation was applied twice a day, 6 days a week, for 2 weeks.

## Range of motion of the ankle joint dorsiflexion

At 1 and 2 weeks after immobilization, the rats were anesthetized, and the range of motion (ROM) on ankle joint dorsiflexion was determined with a goniometer. ROM was measured as the angle (0˚–180˚) between the line connecting the fifth metatarsal to the malleolus lateralis of the fibula and the line connecting the malleolus lateralis of the fibula to the center of the knee joint, and the ankle was passively dorsiflexed with a tension of 0.3 N using a tension gauge (Shiro Industry, Osaka, Japan) [2].

## Tissue sampling and preparation

The left and right soleus muscles of all rats were excised after the experimental period. The right soleus muscles were embedded in tragacanth, and the muscle sample was frozen in liquid

nitrogen. Serial frozen cross sections of the muscles were mounted on glass slides for histological and immunohistochemical analyses. A part of the left soleus muscle was rapidly frozen with liquid nitrogen for biochemical analysis. The remaining left soleus muscles were treated with RNAlater® (Ambion, CA, USA) immediately after excision for use in the molecular biological analysis.

## Histological analysis

Cross sections were stained with hematoxylin and eosin (H&E) stain (Mayer's hemalum solution, Merck KGaA, Darmstadt, Germany; Eosin Y disodium salt, Merck KGaA), picrosirius red stain (Picrosirius red stain kit, Polysciences, PA, US), and ATPase stain (Adenosine 5′-triphosphate disodium salt hydrate, Merck KGaA), each stain method use previous protocols [2, 3, 13]. Then, the dyed cross-sections of muscle were evaluated under an optical microscope. First, H&E-stained cross sections were used to identify the myofiber morphological characteristics and signs of previous muscle injury, such as centralized nuclei. Next, the picrosirius red-stained cross sections were used to identify the perimysium and endomysium in the soleus muscle. Additionally, the ATPase-stained cross-sectional area (CSA) of the myofibers was analyzed using Scion image software (National Institutes of Health, MD, USA). More than 100 myofiber measurements (of type I and II fibers) were recorded per animal [3, 14].

## Immunohistochemical analysis

Cross-sections were air-dried and fixed in ice-cold acetone for 10 min. To inhibit endogenous peroxidase, the sections were incubated with 0.3% $H_2O_2$ in methanol for 40 min at 37°C. After washing with 0.01 M phosphate-buffered saline (PBS; pH 7.4), the sections were incubated for 10 min at 37°C with 0.1% Triton X-100 in PBS. The sections were blocked with 5% bovine serum albumin in PBS for 60 min and incubated overnight at 4°C with a mouse anti-CD-11b primary antibody (1:2000; BMA Biomedicals, Augst, Switzerland) or a rabbit polyclonal anti-dystrophin primary antibody (1:1000; Abcam, Cambridge, UK). The sections were rinsed in PBS for 15 min, and incubated with biotinylated goat anti-mouse IgG (1:1000; Vector Laboratories, CA, USA) or biotinylated goat anti-rabbit IgG (1:1000, Vector Laboratories) for 60 min at 37°C. The sections were then rinsed in PBS and allowed to react with avidin-biotin peroxidase complexes (VECTASTAIN Elite ABC kit, Vector Laboratories) for 60 min at 37°C. Horseradish peroxidase binding sites were visualized with 0.05% 3,3-diaminobenzidine and 0.01% $H_2O_2$ in 0.05 M Tris buffer at 37°C. After a final washing step, the CD-11b sections were covered according to the conventional method, whereas dystrophin sections were covered after staining with hematoxylin. The sections were observed under an optical microscope. Using microscopy and standardized light conditions, the sections were magnified to 400× (CD-11b) or 200× (dystrophin), and images were captured with a digital camera (Nikon, Tokyo, Japan). The number of macrophages was determined from the 400× images by counting the number of CD-11b positive cells per 100 muscle fibers. Vascular areas were omitted from the analysis. Additionally, the number of myonuclei and cross-sectional area were determined from 200× images. To be more specific, myonuclei was measured by counting the nuclei located inside dystrophin and adjacent to myocyte cytoplasm, and the cross-sectional area of the myofibers was measured using Scion image software. Furthermore, the myonuclear domain size, which is defined as the cytoplasmic region controlled by a single myonuclei, was calculated by dividing the cross-sectional area by the number of myonuclei per myofibers [15]. These analyses were conducted using the double-blind method.

**Table 1. Arrangement of synthetic gene-specific primers.**

| Object gene | Arrangement | | Gene Bank No. |
|---|---|---|---|
| | Forward | Reverse | |
| Atrogin-1 | 5'-ACTAAGGAGCGCCATGGATACT-3' | 5'-GTTGAATCTTCTGGAATCCAGGAT-3' | AY059628.1 |
| MuRF-1 | 5'-TGACCAAGGAAAACAGCCACCAG-3' | 5'-TCACTCTTCTTCTCGTCCAGGATGG-3' | AY059627.1 |
| MCP-1 | 5'-CACTCACCTGCTGCTACTCAT-3' | 5'-CTACAGCTTCTTTGGGACACCT-3' | M57441.1 |
| IL-1β | 5'-AATGACCTGTTCTTTGAGGCTGAC-3' | 5'-CGAGATGCTGCTGTGAGATTTGAA-3' | BC091141.1 |
| TGF-β1 | 5'-AGAAGTCACCCGCGTGCTAAT-3' | 5'-CACTGCTTCCCGAATGTCTGA-3' | BC076380.1 |
| α-SMA | 5'-CGGGCTTTGCTGGTGATG-3' | 5'-GGTCAGGATCCCTCTCTTGCT-3' | BC158550.1 |
| β-actin | 5'-GTGCTATGTTGCCCTAGACTTCG-3' | 5'-GATGCCACAGGATTCCATACCC-3' | BC063166.1 |

MuRF, muscle RING-finger protein; MCP, monocyte chemotactic protein; IL, interleukin; TGF, transforming growth factor; SMA, smooth muscle actin.

## Molecular biological analysis

The soleus muscles were used for this analysis. Total RNA was extracted from muscle samples using a RNeasy Fibrous Tissue Mini Kit (Qiagen, CA, USA). Total RNA was used as a template with a QuantiTect® Reverse Transcription Kit (Qiagen) to prepare cDNA, and real-time RT-PCR was performed using Brilliant III Ultra-Fast SYBR Green QPCR Master Mix (Agilent Technologies, CA, USA). The cDNA concentration of all samples was unified to 25 ng/µl, the cDNA was applied 0.2 µl to each well. The synthetic gene-specific primers are listed in Table 1. The threshold cycle (Ct) was determined using an Mx3005P Real-Time QPCR System (Agilent Technologies). The mRNA expression of target genes was calculated using the ΔΔct method.

## Biochemical analysis

The soleus muscles were assessed for hydroxyproline expression using our previous method[3]. Briefly, the muscle samples were immersed in 1.0 M PBS (pH 7.4) and homogenized using Micro Smash™ (MS-100R; Tomy, Tokyo, Japan). Subsequently, the muscle samples were hydrolyzed in 6 N HCl for 15 h at 110°C and then dried in 6 N HCl with an evaporator (EZ-2 HCL-resistant model; Ikeda Scientific, Tokyo, Japan). The muscle samples were hydrolyzed in NAOH for 1 h at 90°C. The hydrolyzed specimens were then mixed with buffered chloramine-T reagent and subsequently oxidized at 20°C. The chromophore was developed by adding Ehr-lich's aldehyde reagent. The absorbance of each sample was measured at 540 nm using Spectra-Max 190 (Molecular Devices, CA, USA). Absorbance values were plotted against the concentration of standard hydroxyproline. The presence of hydroxyproline in the unknown sample extracts was determined from the standard curve. The hydroxyproline concentration of samples was calculated as the content per dry weight (µg/mg dry weight).

## Statistical analysis

All data are presented as the mean ± standard deviation. Differences between groups were assessed using a one-way analysis of variance (ANOVA) followed by Scheffé's method. Differences were considered significant at $p < 0.05$.

## Results

### ROM on ankle joint dorsiflexion

The ROM on dorsiflexion in the control group was 160° at 1 and 2 weeks after immobilization. In the immobilization group, the ROM was 117.1° ± 4.5° at 1 week and 103.8° ± 3.8° at 2

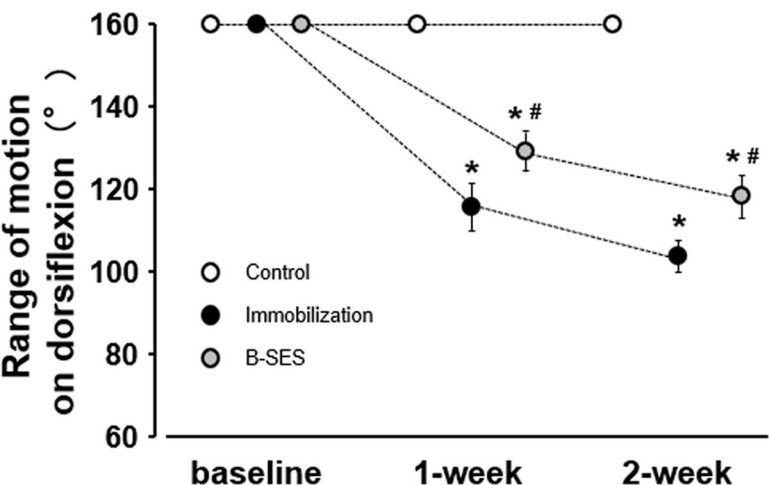

**Fig 2. Range of motion of the ankle joint on dorsiflexion.** Data presented as mean ± standard deviation. *, Significant difference ($p < 0.05$) compared with the control group. #, Significant difference ($p < 0.05$) compared with the immobilization group.

weeks after immobilization. In the B-SES group, the ROM was 132.9˚ ± 2.6˚ at 1 week and 118.3˚ ± 4.9˚ at 2 weeks after immobilization (Fig 2). The ROM of dorsiflexion in the immobilization group and B-SES group was significantly lower than that in the control group and was higher in the B-SES group than in the immobilization group.

## Number of myonuclei and myonuclear domain size

The average number of myonuclei in each myofiber was 2.1 ± 0.2 in the control group at 2 weeks after immobilization. In the immobilization and B-SES groups, the number was 1.5 ± 0.1 and 2.0 ± 0.3, respectively, at 2 weeks after immobilization (Fig 3B). The number of myonuclei was significantly higher in the B-SES group than in the immobilization group, whereas there was no significant difference between the B-SES and control groups.

The myonuclear domain size was 1357.1 ± 200.9 μm$^2$ in the control group. The myonuclear domain size in the immobilization and B-SES groups was 929.1 ± 112.2 and 1009.7 ± 47.3 μm$^2$, respectively, at 2 weeks after immobilization (Fig 3C). The myonuclear domain size in the immobilization and B-SES groups was significantly lower than that in the control group.

## H&E—imaging and cross-sectional area

On the H&E-stained cross sections, except for atrophic changes, abnormal findings were not apparent in the experimental group. On the ATPase-stained cross sections, the CSA of type I myofibers was 2739.5 ± 461.8 μm$^2$ in the control group. The CSA of type I myofiber in the immobilization and B-SES groups was 1429.2 ± 99.7 and 1945.7 ± 300.3 μm$^2$, respectively, at 2 weeks after immobilization (Fig 4B). The CSA of type II myofibers was 1966.2 ± 287.3 μm$^2$ in the control group. The CSA of type II myofiber in the immobilization and B-SES groups was 984.4 ± 183.4 and 1003.3 ± 157.5 μm$^2$, respectively, at 2 weeks after immobilization (Fig 4C). The CSA of type I and II myofibers in the immobilization and B-SES groups was significantly lower than that in the control group, whereas the CSA of type I myofibers in the B-SES group was higher than that in the immobilization group.

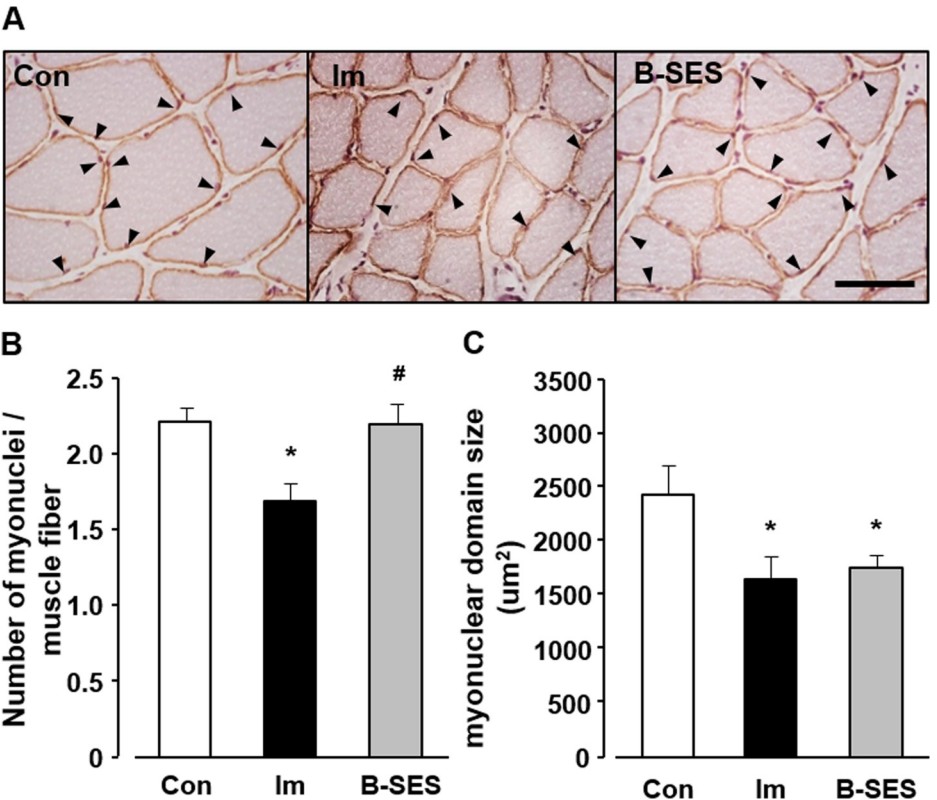

**Fig 3. Number of myonuclei and myonuclear domain in soleus muscles.** (A) Immunohistochemical staining for dystrophin/nuclei of soleus muscles. Arrowheads indicate the myonuclei. Scale bar, 50 μm. (B) Number of myonuclei in each myofiber. (C) myonuclear domain. Open bars represent the control group (Con). Black bars represent the immobilization group (Im). Gray bars represent the belt electrode-skeletal muscle electrical stimulation group (B-SES). Data presented as mean ± standard deviation. *, Significant difference (p < 0.05) compared with the control group. #, Significant difference (p < 0.05) compared with the immobilization group.

## Atrogin-1 and MuRF-1 mRNA expression

Atrogin-1 mRNA expression was 1.0 ± 0.3 in the control group. In the immobilization and B-SES groups, the expression was 4.0 ± 1.0 and 3.9 ± 1.0, respectively, at 2 weeks after immobilization (Fig 5A). MuRF-1 mRNA expression was 0.9 ± 0.3 in the control group. In the immobilization and B-SES groups, the expression was 2.2 ± 0.7 and 2.1 ± 0.7, respectively, at 2 weeks after immobilization (Fig 5B). Atrogin-1 and MuRF-1 mRNA expression in the immobilization and B-SES groups was significantly higher than that in the control group.

## Number of macrophages and MCP-1 mRNA expression

The number of CD-11b-positive cells per 100 myofibers was 10.0 ± 2.1 in the control group. In the immobilization and B-SES groups, the number was 31.9 ± 5.1 and 13.4 ± 3.5, respectively, at 2 weeks after immobilization (Fig 6B). MCP-1 mRNA expression was 0.5 ± 0.3 in the control group. In the immobilization and B-SES groups, the expression was 3.2 ± 1.5 and 1.0 ± 0.6, respectively, at 2 weeks after immobilization (Fig 6C). The number of macrophages and MCP-1 mRNA expression was significantly lower in the B-SES group than in the immobilization group, whereas there were no significant differences in these parameters between the B-SES and control groups.

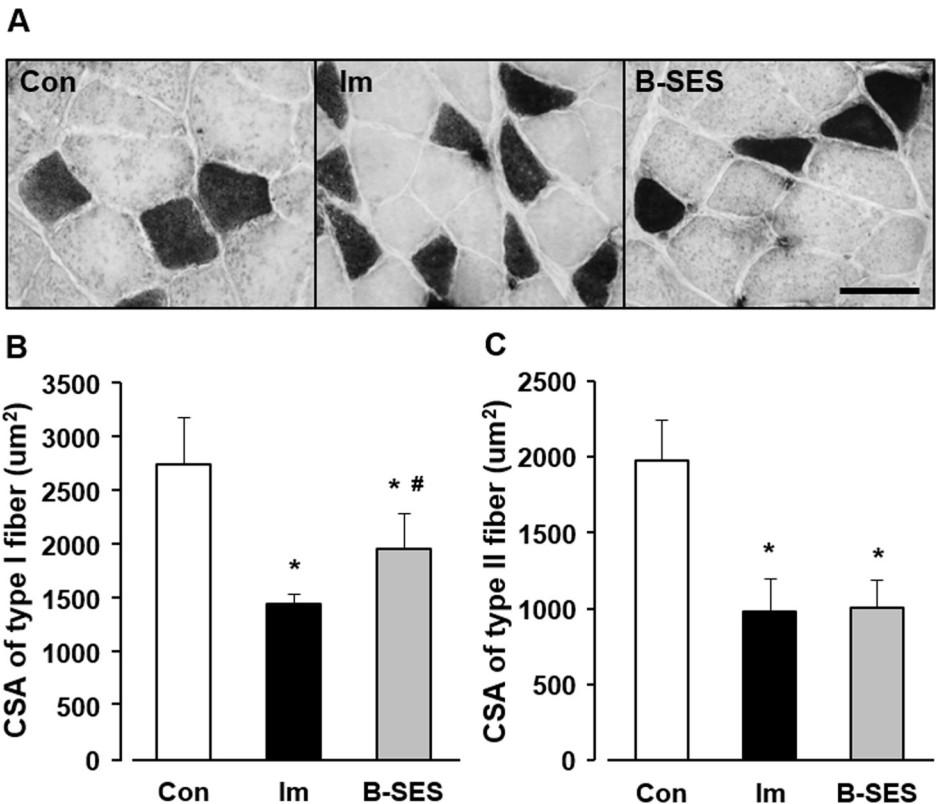

**Fig 4. Cross-sectional area of type I and II fibers in soleus muscles.** (A) ATPase staining of soleus muscles. The white areas indicate type I fibers, whereas the black areas indicate type II fibers. Scale bar, 50 μm. (B) Cross-sectional area (CSA) of type I fibers. (C) CSA of type II fibers. Open bars represent the control group (Con). Black bars represent the immobilization group (Im). Gray bars represent the belt electrode-skeletal muscle electrical stimulation group (B-SES). Data presented as mean ± standard deviation. *, Significant difference ($p < 0.05$) compared with the control group. #, Significant difference ($p < 0.05$) compared with the immobilization group.

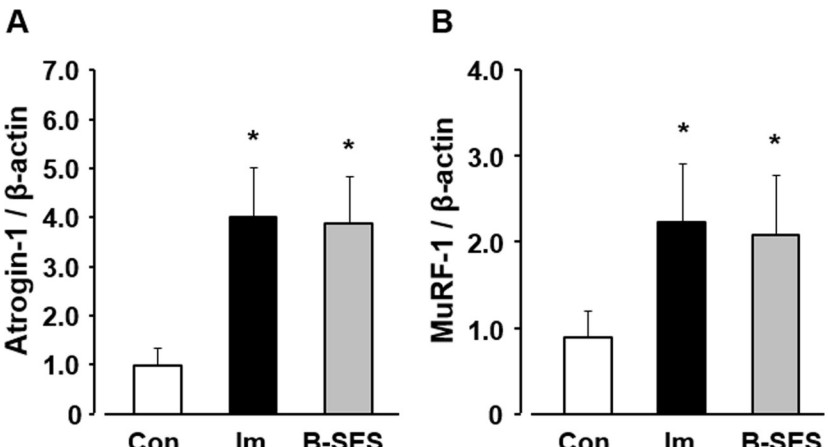

**Fig 5. mRNA expression of Atrogin-1 (A) and MuRF-1 (B) in soleus muscles.** Open bars represent the control group (Con). Black bars represent the immobilization group (Im). Gray bars represent the belt electrode-skeletal muscle electrical stimulation group (B-SES). Data presented as mean ± standard deviation. *, Significant difference ($p < 0.05$) compared with the control group.

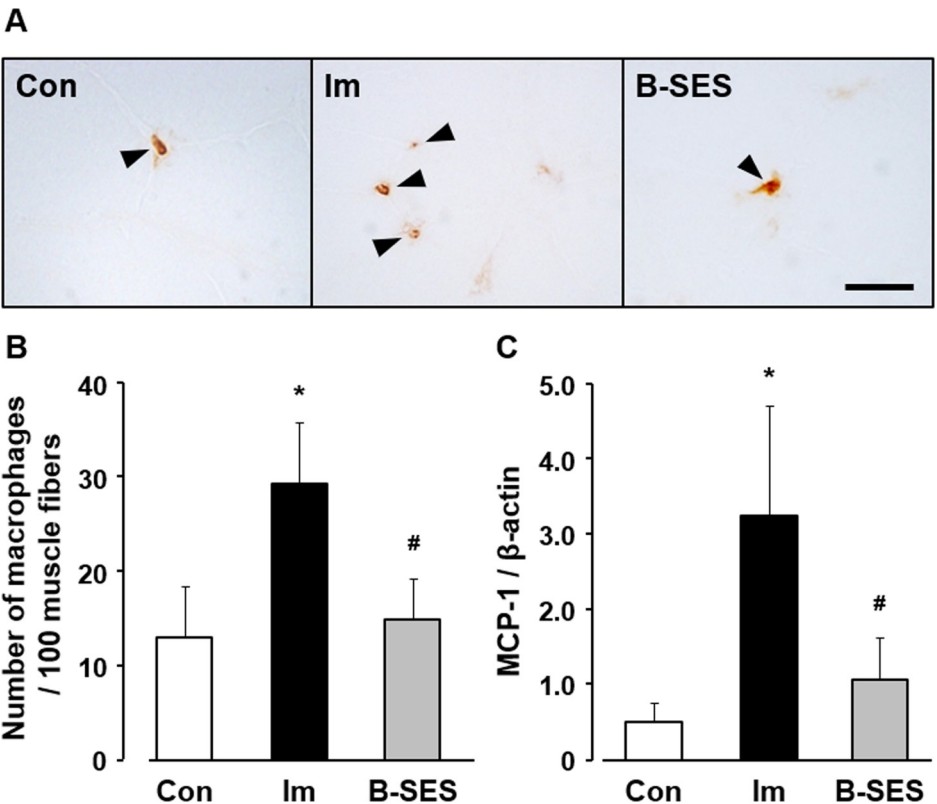

**Fig 6. Macrophages number and MCP-1 mRNA expression in soleus muscles.** (A) Immunohistochemical staining for CD-11b in soleus muscles. Arrowheads indicate the CD-11b-positive cells. Scale bar, 20 μm. (B) Number of CD-11b-positive cells per 100 myofibers. (C) Relative expression of MCP-1 mRNA. Open bars represent the control group (Con). Black bars represent the immobilization group (Im). Gray bars represent the belt electrode-skeletal muscle electrical stimulation group (B-SES). Data presented as mean ± standard deviation. *, Significant difference ($p < 0.05$) compared with the control group.

## IL-1β, TGF-β1, and α-SMA mRNA expression

IL-1β mRNA expression was 1.0 ± 0.4 in the control group. In the immobilization and B-SES groups, the expression was 4.5 ± 1.9 and 1.7 ± 0.5, respectively, at 2 weeks after immobilization (Fig 7A). The TGF-β1 mRNA expression was 0.8 ± 0.3 in the control group. In the immobilization and B-SES groups, the expression was 1.8 ± 0.4 and 0.9 ± 0.3, respectively, at 2 weeks after immobilization (Fig 7B). α-SMA mRNA expression was 0.5 ± 0.1 in the control group. In the immobilization and B-SES groups, the expression was 4.0 ± 1.3 and 0.8 ± 0.2, respectively, at 2 weeks after immobilization (Fig 7C). The IL-1β, TGF-β1, and α-SMA mRNA expression levels were significantly lower in the B-SES group than in the immobilization group, and these parameters were not significantly different between the B-SES and control groups.

## Picrosirius red imaging and hydroxyproline expression

An evaluation of the picrosirius red images demonstrated that the perimysium and endomysium were thicker in the immobilization group than in the control and B-SES groups (Fig 8A). The expression of hydroxyproline was 3.3 ± 0.7 μg/mg dry weight in the control group at 2 weeks after immobilization. In the immobilization and B-SES groups, the expression was 7.1 ± 1.3 and 4.1 ± 0.8 μg/mg dry weight, respectively, at 2 weeks after immobilization (Fig 8B). The level of hydroxyproline expression was significantly lower in the B-SES group than in

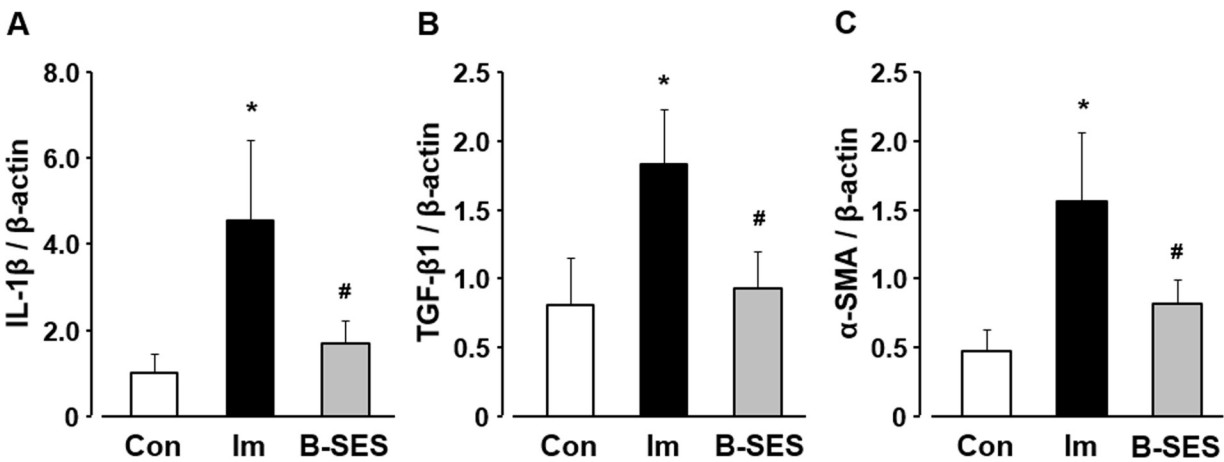

**Fig 7. mRNA expression of IL-1β (A), TGF-β1 (B), and α-SMA (C) in soleus muscles.** Open bars represent the control group (Con). Black bars represent the immobilization group (Im). Gray bars represent the belt electrode-skeletal muscle electrical stimulation group (B-SES). Data presented as mean ± standard deviation. *, Significant difference ($p < 0.05$) compared with the control group. #, Significant difference ($p < 0.05$) compared with the immobilization group.

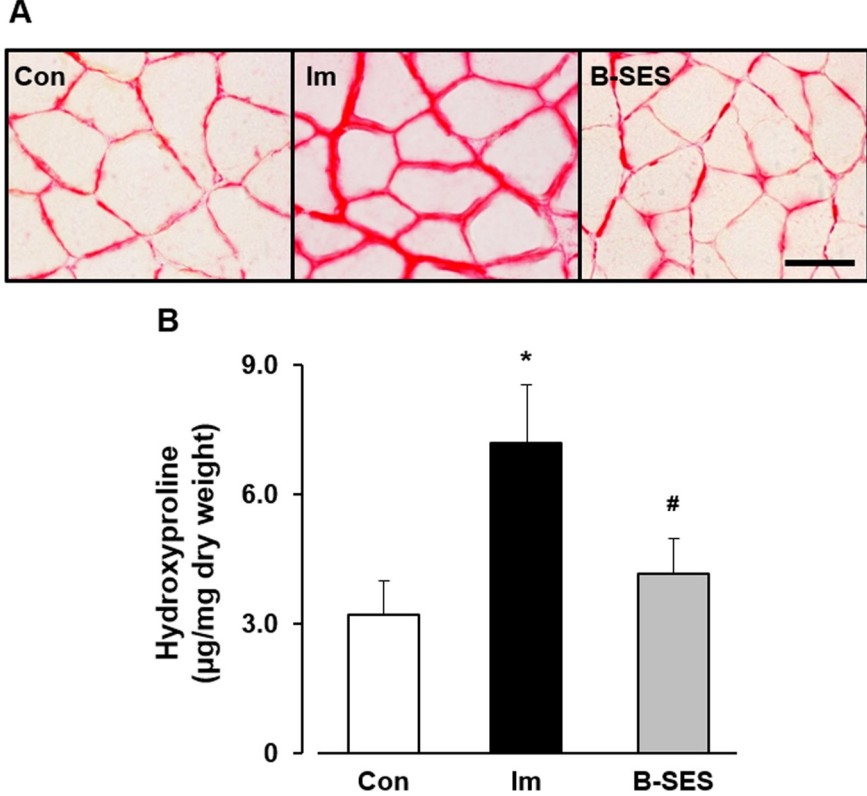

**Fig 8. Picrosirius red imaging(A) and hydroxyproline expression (B) in soleus muscles.** Scale bar, 50 μm. Open bar represents the control group (Con). Black bar represents the immobilization group (Im). Gray bar represents the belt electrode-skeletal muscle electrical stimulation group (B-SES). Data presented as mean ± standard deviation. *, Significant difference ($p < 0.05$) compared with the control group. #, Significant difference ($p < 0.05$) compared with the immobilization group.

the immobilization group, and this parameter was not significantly different between the B-SES and control groups.

## Discussion

This research investigated the biological effect of B-SES on immobilization-induced muscle fibrosis with histological, immunohistochemical, biochemical, and molecular biological analyses.

In the immobilization group, the ROM on dorsiflexion decreased to 73.2% and 64.9% of that in the control group at 1 and 2 weeks after immobilization, consistent with the results of previous studies [2, 3]. These ROM results demonstrated the occurrence of muscle contracture in the immobilization group; thus, the results show that the immobilization treatment used in this study was appropriate.

The number of myonuclei, the myonuclear domain size, and the CSA of type I and II myofibers in the immobilization group was significantly lower than that in the control group. Myofibers are multinucleated cells, and this feature has led to the myonuclear domain theory, which describes the theoretical amount of cytoplasm within a myofiber that can be regulated by a single myonucleus [6]. Furthermore, the defining features of apoptosis are consistent in most cell types, and cultured myocytes exhibit hallmark morphological and biochemical characteristics in response to apoptotic stimulation [16]. In fact, some previous reports indicated that apoptosis of myonuclei is related to muscle atrophy [4, 5]. Therefore, the decrease in myonuclei might be induced by apoptotic changes in the immobilization group in this study. Next, Atrogin-1 and MuRF-1 mRNA expression increased in immobilized rat soleus muscles. Another study demonstrated that Atrogin-1 and MuRF-1 as ubiquitin-proteasome pathway enzymes were upregulated in immobilized skeletal muscles [17]. Consequently, our research surmised that type I and II myofiber atrophy is caused by the loss of myonuclei and the upregulation of degrading system of muscular protein.

From the analyses of macrophages and MCP-1, our study outcomes revealed that the number of macrophages and MCP-1 mRNA expression increased in immobilized rat soleus muscles. Macrophages are characterized by avid phagocytosis, as macrophages phagocytose unnecessary skeletal muscle cytoplasm during muscle regeneration [18]. Moreover, apoptotic bodies are formed around unnecessary cytoplasmic regions formed by apoptosis of myonuclei, and clearance of skeletal muscle is promoted by phagocytosis of apoptotic bodies by macrophages [7]. Therefore, this study inferred that macrophages accumulated to phagocytose the unnecessary cytoplasm resulting from muscle atrophy.

Previous research has shown that macrophages are the main source of IL-1β in fibrotic lesions (macrophages can induce IL-1β production) [19]. The present study showed that IL-1β mRNA expression increased in the soleus muscle after 2 weeks of immobilization. Our previous study showed that the increase in macrophages was related to the upregulation of IL-1β mRNA in immobilized skeletal muscle [3]. IL-1β is a potent inducer of TGF-β1 synthesis, which induces fibroblast activation and collagen production in a TGF-β-dependent manner [20]. Several reports have shown that TGF-β1 is a key player in fibrotic diseases [21] and an important component of muscle fibrosis [22]. Moreover, TGF-β1 is a key factor that promotes the conversion of fibroblasts into myofibroblasts, a differentiation process commonly associated with pathological conditions such as fibrosis [23]. Myofibroblasts produce large amounts of collagen [24] and play a major role in pathological contractures, including Dupuytren contracture, plantar fibromatosis, and frozen shoulder [23]. In this study, TGF-β1 and α-SMA mRNA expression and hydroxyproline expression increased in the rat soleus muscle at 2 weeks after immobilization. Additionally, picrosirius red images showed that the perimysium

and endomysium were thicker in the immobilization group than in the control group. These results indicate that TGF-β1 mRNA upregulation affects the differentiation of fibroblasts into myofibroblasts during the early stage of immobilization, and that these alterations are associated with the incidence of immobilization-induced muscle fibrosis. To summarize, in the immobilization group, macrophages accumulate to phagocytose unnecessary muscle cytoplasm, and IL-1β/TGF-β1 signaling via macrophage accumulation affects the incidence of immobilization-induced muscle fibrosis in muscle contracture.

B-SES can supply electrical stimulation to widespread lower-limb skeletal muscles, this intervention has a high efficiency in contracting the skeletal muscle. Furthermore, a preliminary experiment in our laboratory showed that B-SES was less likely to cause muscle fatigue but can still provide enough electrical stimulation for contractile exercise of skeletal muscle. Therefore, B-SES was used as the electrical stimulation method in the present study, and muscle contractile exercise was performed on the soleus muscle. The results showed that the number of myonuclei in the B-SES group was not significantly different from that in the control group. In a review on the effect of electrical stimulation on preventing myonuclei decrease, the mechanical stimuli by electrical stimulation decreased apoptosis by activating several signaling pathways (e.g., mitogen-activated protein kinase pathway) within the cells [25]. Additionally, a pilot study demonstrated that electrical stimulation reduced cell apoptosis via regulation of pro- and anti-apoptotic proteins [26, 27]. Our results may suggest that muscle contractile exercise through B-SES prevented the decrease in myonuclei induced by apoptotic changes. However, the myonuclear domain size, Atrogin-1 and MuRF-1 levels did not change with muscle contractile exercise by B-SES. Myofiber atrophy was caused by the loss of myonuclei and the upregulation of degrading system of muscular protein. From our results, the muscle contractile exercise through B-SES had no effect for mitigating the upregulation of degrading system of muscular protein, despite preventing the decrease in myonuclei. Therefore, we surmised that mitigation of decreasing myonuclei via B-SES led to suppress the partial type I myofiber atrophy.

The increase in macrophage number and the upregulation of MCP-1 were suppressed in the B-SES group. Macrophages were divided into M1 macrophages (inflammatory macrophages) and M2 macrophages (anti-inflammatory macrophages), M1 macrophages played a key role of fibrous lesion via IL-1β/TGF-β1 signaling. In our previous study, M1 macrophages increased in 2-week immobilized rat skeletal muscles [28], we surmised that macrophages accumulation was occurred by M1 macrophages increase in immobilized rat soleus muscles. Also, other previous report showed that physical exercise reduced the M1 macrophage response in myocardial ischemic injury [29]. Namely, the muscle contractile exercise through B-SES might be effective for the reducing M1 macrophage. Moreover, the upregulation of IL-1β, TGF-β1, and α-SMA mRNA and collagen content induced by immobilization were suppressed in the B-SES group. A previous study indicated that IL-1β expression in skeletal muscle was significantly lower in the exercise group than in the sedentary group [30]. Furthermore, Blaauboer et al. demonstrated that mechanical stimulation of human lung fibroblasts inhibited TGF-β1, α-SMA, and type I and III collagen mRNA expression [24]. Another study showed that treadmill exercise in mice attenuated TGF-β1 protein expression and collagen deposition in the gastrocnemius-soleus muscle [31]. Additionally, cyclic muscle contraction induced by electrical stimulation reduced TGF-β1, α-SMA, and type I and III collagen mRNA expression in immobilized rat soleus muscle, and mechanical stimulation played a key role in the regulation of fibrosis-related factors [7]. Based on the available literature and our results, the effect of muscle contractile exercise through B-SES on fibrosis-related factors may be due to the mechanisms described in these reports. Namely, the muscle contractile exercise through B-SES mitigated the macrophages accumulation via the myonuclei decrease, these

alterations might prevent the differentiation of fibroblasts into myofibroblasts via IL-1β/TGF-β1 signaling. And, this research surmised that these biological regulations led to suppress immobilization-induced muscle fibrosis.

This study has several limitations. First, it is uncertain whether the current electrical stimulation protocol is the most effective interventions. Further examination of various frequencies, intensities, duty cycles, times, and intraday sessions of electrical stimulation protocols are required. Additionally, this study was unable to determine why the muscle contractile exercise through B-SES only showed an effect on type I fibers of soleus muscles. Further studies on the detailed changes in type I and II fibers in immobilized soleus muscle are needed to address this issue. Moreover, the present study could not confirm whether myonuclear apoptosis can be suppressed by contractile exercise of skeletal muscle through B-SES. Future studies using TUNEL staining of muscle sections are needed to answer this question. Finally, data related to the cause-and-effect relationship of cellular and molecular events are insufficient in our study. Thus, future studies using an antagonist or inhibitor are warranted to address this limitation.

In summary, muscle contractile exercise through B-SES may prevent immobilization-induced muscle fibrosis, and this alteration may suppress the development of limited ROM on dorsiflexion. Therefore, our study surmised that muscle contractile exercise through B-SES may be effective against muscle contracture.

## Supporting information

**S1 File.**
(XLSX)

## Acknowledgments

The belt electrode-skeletal muscle electrical stimulation system was provided by HOMER ION Co., Ltd. (Tokyo, Japan). We would like to thank Editage (www.editage.com) for English language editing.

## Author Contributions

**Conceptualization:** Yuichiro Honda, Yasutaka Kondo, Hideki Kataoka, Junya Sakamoto, Minoru Okita.

**Data curation:** Yuichiro Honda.

**Formal analysis:** Yuichiro Honda, Yasutaka Kondo, Hideki Kataoka, Junya Sakamoto.

**Funding acquisition:** Yuichiro Honda, Minoru Okita.

**Investigation:** Yuichiro Honda, Natsumi Tanaka, Yasuhiro Kajiwara.

**Methodology:** Yuichiro Honda.

**Project administration:** Yuichiro Honda, Minoru Okita.

**Resources:** Yuichiro Honda, Ryuji Akimoto, Atsushi Nawata.

**Supervision:** Minoru Okita.

**Validation:** Yuichiro Honda, Natsumi Tanaka, Yasuhiro Kajiwara.

**Visualization:** Yuichiro Honda.

**Writing – original draft:** Yuichiro Honda.

**Writing – review & editing:** Yuichiro Honda.

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
