## [Decision Letter · Decision Letter 0]

11 Feb 2021

PONE-D-20-37553

Effect of belt electrode-skeletal muscle electrical stimulation on immobilization-induced muscle fibrosis

PLOS ONE

Dear Dr. Honda,

Thank you for submitting your manuscript to PLOS ONE. After careful consideration, we feel that it has merit but does not fully meet PLOS ONE’s publication criteria as it currently stands. Therefore, we invite you to submit a revised version of the manuscript that addresses the points raised during the review process.

Specifically, parts of the manuscript needs to be re-written to address the reviewer concerns. Concerns with Figures 2, 5, and 6 also needs to be addressed.

We look forward to receiving your revised manuscript.

Kind regards,

Aldrin V. Gomes, Ph.D.

Academic Editor

PLOS ONE

Journal Requirements:

2. Thank you for stating the following in the Financial Disclosure* (delete as necessary) section:

"No.2

● Grant name awarded to each author: research funding from ALCARE Co., Ltd.

● The full name of each funder: ALCARE Co., Ltd.

● URL of each funder website: https://www.alcare.co.jp/

● Did the sponsors or funders play any role in the study design, data collection and analysis, decision to publish, or preparation of the manuscript?: Include this sentence at the end of your statement: The funders had no role in study design, data collection and analysis, decision to publish, or preparation of the manuscript."

We note that one or more of the authors are employed by a commercial company: ALCARE Co., Ltd, HOMER ION Co., Ltd.

2.1. Please provide an amended Funding Statement declaring this commercial affiliation, as well as a statement regarding the Role of Funders in your study. If the funding organization did not play a role in the study design, data collection and analysis, decision to publish, or preparation of the manuscript and only provided financial support in the form of authors' salaries and/or research materials, please review your statements relating to the author contributions, and ensure you have specifically and accurately indicated the role(s) that these authors had in your study. You can update author roles in the Author Contributions section of the online submission form.

2.2. Please also provide an updated Competing Interests Statement declaring this commercial affiliation along with any other relevant declarations relating to employment, consultancy, patents, products in development, or marketed products, etc.  

Reviewers' comments:

Reviewer's Responses to Questions

**Comments to the Author**

1. Is the manuscript technically sound, and do the data support the conclusions?

Reviewer #1: Partly

2. Has the statistical analysis been performed appropriately and rigorously? 

Reviewer #1: Yes

3. Have the authors made all data underlying the findings in their manuscript fully available?

Reviewer #1: Yes

4. Is the manuscript presented in an intelligible fashion and written in standard English?

Reviewer #1: Yes

5. Review Comments to the Author

Reviewer #1: This manuscript describes the effects of belt electrode skeletal muscle electrical stimulation in a rodent model of immobilization-induced muscle atrophy. The authors found that belt electrode skeletal muscle electrical stimulation decreased markers of inflammation mediated by macrophages but did not decrease markers of atrophy. Further, the authors found that the belt electrode skeletal muscle electrical stimulation improved range of motion. The data have the potential to add new information to the field. Below are comments that require attention and can help strengthen and improve the manuscript.

Methods

General comment

The study could benefit from having a control groups for BES.

Specific comments

Please list the ages of the rats

Please list whether rats were individually, or group housed

Please list the reagents/kits used to perform the stains. Or at the very least, reference a paper indicating the use of those stains in your lab or other

Please indicate whether primer pairs were validated with the housekeeping gene (e.g., did gene of interest and housekeeping gene amplify targets with similar efficiency).

Please indicate the amount of cDNA used for amplification.

Please indicate whether PCR products were confirmed to be specific by sequencing.

Results

Specific comments

Please define ROM

Figure 2 suggests that the density of myonuclei is greater in BES group vs immobilization or control (same number of nuclei compared to control, but decreased area; greater number of nuclei compared to immobilization but same area as immobilization). This suggest that BES is decreasing atrophy of the muscle fibers. However, in Figure 5, two markers of atrophy are not decreased by BES. Can the authors please comment in the discussion regarding this observation and how they are compatible with one another?

For Figure 6C, please provide rationale for measuring 100 fibers per group. For example, provide a reference or rationale indicating that it either the standard or allows for adequate sampling.

Discussion

General comments

Some studies suggest that macrophages are important for promoting recovery from atrophy. Please include a few sentences or paragraph concerning how the current findings fit within data suggesting a protective role of macrophages.

Specific comments

Lines 397-400, page 18, please check this sentence as it is can be improved on clarity. “Namely, the macrophages accumulation via the myonuclei decrease was suppressed by muscle contractile exercise through B-SES, our investigation surmised that this alteration led to the prevention of immobilization-induced muscle fibrosis via the suppression of fibrosis related factor”

6. PLOS authors have the option to publish the peer review history of their article (what does this mean?). If published, this will include your full peer review and any attached files.

Reviewer #1: No

---

## [Author Response · Author response to Decision Letter 0]

20 Feb 2021

Response to the Editor-in-Chief and reviewer

We would like to thank both the reviewers for the evaluation of our study, as well as their constructive criticism, which allowed us to strengthen and clarify our study's conclusions. We have responded in detail to each comment and discussed how the concerns raised were addressed in the revised manuscript.

Journal Requirements

1. Please ensure that your manuscript meets PLOS ONE's style requirements, including those for file naming. The PLOS ONE style templates can be found at https://journals.plos.org/plosone/s/file?id=wjVg/PLOSOne_formatting_sample_main_body.pdf and https://journals.plos.org/plosone/s/file?id=ba62/PLOSOne_formatting_sample _title_authors_affiliations.pdf

We rechecked our manuscript according to The PLOS ONE style templates.

 

2.1. Please provide an amended Funding Statement declaring this commercial affiliation, as well as a statement regarding the Role of Funders in your study. If the funding organization did not play a role in the study design, data collection and analysis, decision to publish, or preparation of the manuscript and only provided financial support in the form of authors' salaries and/or research materials, please review your statements relating to the author contributions, and ensure you have specifically and accurately indicated the role(s) that these authors had in your study. You can update author roles in the Author Contributions section of the online submission form. 

We updated author roles of “Atsushi Nawata” and “Ryuji Akimoto” to “Resources”. Additionally, please add following statement to our amended Funding Statement:

“The funder provided support in the form of salaries for Akimoto R and Nawata A, but did not have any additional role in the study design, data collection and analysis, decision to publish, or preparation of the manuscript. The specific roles of these authors are articulated in the ‘author contributions’ section.”

2.2. Please also provide an updated Competing Interests Statement declaring this commercial affiliation along with any other relevant declarations relating to employment, consultancy, patents, products in development, or marketed products, etc. 

We confirmed that this commercial affiliation does not alter our adherence to all PLOS ONE policies on sharing data and materials by including the following. Please add following statement to our Competing Interests Statement:

“This does not alter our adherence to PLOS ONE policies on sharing data and materials.”

 

Reviewer: 1

1. The study could benefit from having a control groups for B-SES.

Your advice is very important. However, the Ethics Review Committee for Animal Experimentation of Nagasaki University warned this study to use the minimum number of rats required. Therefore, we could not add the control + B-SES group in this research. Presently, we are considering this point in another project.

2. Please list the ages of the rats

We used “Eight-week-old male Wistar rats” (page 6, line 109).

3. Please list whether rats were individually, or group housed.

Two rats were housed in each cage (page 6, line 111).

4. Please list the reagents/kits used to perform the stains. Or at the very least, reference a paper indicating the use of those stains in your lab or other

We added the reagents/kits used to each staining and references related to stain protocols (page 9, line 172-176). Also, we added following research in reference section (page 21, line 457-459).

14. Hintz CS, Coyle EF, Kaiser KK, Chi MM, Lowry OH. Comparison of muscle fiber typing by quantitative enzyme assays and by myosin ATPase staining. J Histochem Cytochem 1984 ;32: 655-660.

5. Please indicate whether primer pairs were validated with the housekeeping gene (e.g., did gene of interest and housekeeping gene amplify targets with similar efficiency).

We confirmed amplification efficiency of all primer pairs in pilot study. In addition, the amplification efficiency of all primer pairs was 90-110%. From these data, we considered that the gene of interest and housekeeping gene amplified targets with similar efficiency.

6. Please indicate the amount of cDNA used for amplification.

The cDNA concentration of all samples was unified to 25 ng/μl. And, we applied 0.2 μl cDNA to each well.

7. Please indicate whether PCR products were confirmed to be specific by sequencing.

We could not research the PCR products by the sequencer device. However, we confirmed the specification of PCR products with Basic Local Alignment Search Tool (BLAST, NIH, US). All PCR products were indicated high percent identify and low E-value, no sequence of molecule different from the target was observed.

8. Please define ROM

We described definition of ROM according your advice (page 8, line 157-162).

 

9. Figure 2 suggests that the density of myonuclei is greater in BES group vs immobilization or control (same number of nuclei compared to control, but decreased area; greater number of nuclei compared to immobilization but same area as immobilization). This suggest that BES is decreasing atrophy of the muscle fibers. However, in Figure 5, two markers of atrophy are not decreased by BES. Can the authors please comment in the discussion regarding this observation and how they are compatible with one another?

Thank you for your important suggestion. The number of myonuclei was significantly higher in the B-SES group than in the immobilization group, whereas there was no significant difference between the B-SES and control groups. However, the myonuclear domain size, which is defined as the cytoplasmic region controlled by a single myonuclei, and Atrogin-1/MuRF-1 mRNA expression in Immobilization and B-SES groups were no significant difference. In summary, the muscle contractile exercise through B-SES may prevent the decreasing myonuclei, but not may be possible to suppress the decrease in muscular protein produced by single myonuclei.

Myofiber atrophy was caused by the loss of myonuclei and the upregulation of degrading system of muscular protein. From our results, the muscle contractile exercise through B-SES had no effect for mitigating the upregulation of degrading system of muscular protein, despite the preventing the decrease in myonuclei. Therefore, we surmised that mitigation of decreasing myonuclei via B-SES led to suppress the partial type I myofiber atrophy. Therefore, we revised sentence according the above contents (page 17, line 372- page 18, line 378).

 

10. For Figure 6C, please provide rationale for measuring 100 fibers per group. For example, provide a reference or rationale indicating that it either the standard or allows for adequate sampling.

Our laboratory presented the data by this method in previous works (following articles). Therefore, we added reference 3 and 13 to page 9, line 183.

3. Honda Y, Sakamoto J, Nakano J, Kataoka H, Sasabe R, Goto K, et al. Upregulation of interleukin-1beta/transforming growth factor-beta1 and hypoxia relate to molecular mechanisms underlying immobilization-induced muscle contracture. Muscle Nerve 2015;52: 419–427.

13. Matsumoto Y, Nakano J, Oga S, Kataoka H, Honda Y, Sakamoto J, et al. The non-thermal effects of pulsed ultrasound irradiation on the development of disuse muscle atrophy in rat gastrocnemius muscle. Ultrasound Med Biol 2014;40: 1578-1586.

11. Some studies suggest that macrophages are important for promoting recovery from atrophy. Please include a few sentences or paragraph concerning how the current findings fit within data suggesting a protective role of macrophages.

We added new sentences according your suggestion (page 18, line 381-385).

12. Lines 397-400, page 18, please check this sentence as it is can be improved on clarity. “Namely, the macrophages accumulation via the myonuclei decrease was suppressed by muscle contractile exercise through B-SES, our investigation surmised that this alteration led to the prevention of immobilization-induced muscle fibrosis via the suppression of fibrosis related factor”

We revised this sentence according your suggestion (page 18, line 397- page 19, line 401).

---

## [Decision Letter · Decision Letter 1]

11 Mar 2021

PONE-D-20-37553R1

Effect of belt electrode-skeletal muscle electrical stimulation on immobilization-induced muscle fibrosis

PLOS ONE

Dear Dr. Honda,

Thank you for submitting your manuscript to PLOS ONE. After careful consideration, we feel that it has merit but does not fully meet PLOS ONE’s publication criteria as it currently stands. Therefore, we invite you to submit a revised version of the manuscript that addresses the points raised during the review process.

Specifically, some of the new information added needs to be re-written and the other minor concerns of the reviewer needs to be addressed.

We look forward to receiving your revised manuscript.

Kind regards,

Aldrin V. Gomes, Ph.D.

Academic Editor

PLOS ONE

Journal Requirements:

Reviewers' comments:

Reviewer's Responses to Questions

**Comments to the Author**

1. If the authors have adequately addressed your comments raised in a previous round of review and you feel that this manuscript is now acceptable for publication, you may indicate that here to bypass the “Comments to the Author” section, enter your conflict of interest statement in the “Confidential to Editor” section, and submit your "Accept" recommendation.

Reviewer #1: (No Response)

2. Is the manuscript technically sound, and do the data support the conclusions?

Reviewer #1: Yes

3. Has the statistical analysis been performed appropriately and rigorously? 

Reviewer #1: Yes

4. Have the authors made all data underlying the findings in their manuscript fully available?

Reviewer #1: Yes

5. Is the manuscript presented in an intelligible fashion and written in standard English?

Reviewer #1: Yes

6. Review Comments to the Author

Reviewer #1: The revised manuscript is significantly improved.

There are just a few minor points below that are recommended for further revision.

Discussion.

The new paragraph on pages 17 and 18 regarding muscle nuclei is not clear.

I recommend stating what you did in your response. So eliminate the highlighted lines 372-378 on the redlined manuscript and replace with below (your response).

“Myofiber atrophy was caused by the loss of myonuclei and the upregulation of degrading system of muscular protein. From our results, the muscle contractile exercise through B-SES had no effect for mitigating the upregulation of degrading system of muscular protein, despite the preventing the decrease in myonuclei. Therefore, we surmised that mitigation of decreasing myonuclei via B-SES led to suppress the partial type I myofiber atrophy”

The paragraphs regarding macrophages do not address the point raised (page 18, lines 381- 385). If macrophages may be beneficial, then B-SES might not be a good long-term for preventing muscle atrophy. Meaning, you are preventing macrophage accumulation, yet some studies suggest that macrophages are important for preventing atrophy. This was what I wanted you to address.

See:

https://www.ncbi.nlm.nih.gov/pmc/articles/PMC2861088/

https://www.ncbi.nlm.nih.gov/pmc/articles/PMC6664695/

Please include in the methods the amount of cDNA added to each well. Although the authors stated it in the response, which I appreciate, the point was to include that information in the manuscript so that others can better reproduce the data.

7. PLOS authors have the option to publish the peer review history of their article (what does this mean?). If published, this will include your full peer review and any attached files.

Reviewer #1: No

---

## [Author Response · Author response to Decision Letter 1]

21 Mar 2021

Response to reviewer

We would like to thank the reviewer for the evaluation of our study, as well as constructive criticism, which allowed us to strengthen and clarify our study's conclusions. We have responded in detail to each comment and discussed how the concerns raised were addressed in the revised manuscript.

Reviewer: 

● I recommend stating what you did in your response. So eliminate the highlighted lines 372-378 on the redlined manuscript and replace with below (your response). “Myofiber atrophy was caused by the loss of myonuclei and the upregulation of degrading system of muscular protein. From our results, the muscle contractile exercise through B-SES had no effect for mitigating the upregulation of degrading system of muscular protein, despite the preventing the decrease in myonuclei. Therefore, we surmised that mitigation of decreasing myonuclei via B-SES led to suppress the partial type I myofiber atrophy”

Following your advice, we replaced this sentence to your recommendation (page 17-18, line 374-379).

● The paragraphs regarding macrophages do not address the point raised (page 18, lines 381- 385). If macrophages may be beneficial, then B-SES might not be a good long-term for preventing muscle atrophy. Meaning, you are preventing macrophage accumulation, yet some studies suggest that macrophages are important for preventing atrophy. This was what I wanted you to address.

We apologized for wronging the mean of your question.

Macrophages were divided into M1 macrophages (inflammatory macrophages) and M2 macrophages (anti-inflammatory macrophages). M1 macrophages related to the proliferation of myoblasts, M2 macrophages matured myoblasts to myotube. Therefore, some researches with reload animal models indicated that M1 macrophages accumulation was beneficial for muscle hypertrophy. However, some researches for fibrous lesions demonstrated that M1 macrophages played a key role of fibrosis via IL-1β/TGF-β1 signaling. 

In our previous study, although M1 macrophages increased in 2-week immobilized rat skeletal muscles (Oga S, Honda Y, et al. Muscle Nerve 61(5): 662-670, 2020), the muscle atrophy occurred in same muscles. In this study, M2 macrophages did not increase even though M1 macrophages increased. We surmised that M1 macrophages did not work for preventing muscle atrophy in this immobilization model. On the other hand, immobilization-induced muscle fibrosis occurred in same immobilized rat skeletal muscles, Additionally, the expression of IL-1β/TGF-β1 mRNA were upregulated (unpublished data). Therefore, M1 macrophages might relate to promote immobilization-induced muscle fibrosis, we surmised that the inhibition of M1 macrophages accumulation via B-SES led to prevent immobilization-induced muscle fibrosis.

Since this investigation was examined for the effect of B-SES to immobilization-induced muscle fibrosis, we added the above content about fibrosis to “Discussion” (page 18, line 381-388). And, we added following reports to “References” (page 23, line 502-503).

28.　Oga S, Goto K, Sakamoto J, Honda Y, Sasaki R, Ishikawa K, et al. Mechanisms underlying immobilization-induced muscle pain in rats. Muscle Nerve 2020;61: 662-670.

● Please include in the methods the amount of cDNA added to each well. Although the authors stated it in the response, which I appreciate, the point was to include that information in the manuscript so that others can better reproduce the data.

We described the amount of cDNA added to each well according your advice (page 11, line 216-217).

---

## [Decision Letter · Decision Letter 2]

21 Apr 2021

Effect of belt electrode skeletal muscle electrical stimulation on immobilization-induced muscle fibrosis

PONE-D-20-37553R2

Dear Dr. Honda,

We’re pleased to inform you that your manuscript has been judged scientifically suitable for publication and will be formally accepted for publication once it meets all outstanding technical requirements.

Kind regards,

Aldrin V. Gomes, Ph.D.

Academic Editor

PLOS ONE

Additional Editor Comments (optional):

Reviewers' comments:

Reviewer's Responses to Questions

**Comments to the Author**

1. If the authors have adequately addressed your comments raised in a previous round of review and you feel that this manuscript is now acceptable for publication, you may indicate that here to bypass the “Comments to the Author” section, enter your conflict of interest statement in the “Confidential to Editor” section, and submit your "Accept" recommendation.

Reviewer #1: All comments have been addressed

2. Is the manuscript technically sound, and do the data support the conclusions?

Reviewer #1: Yes

3. Has the statistical analysis been performed appropriately and rigorously? 

Reviewer #1: Yes

4. Have the authors made all data underlying the findings in their manuscript fully available?

Reviewer #1: Yes

5. Is the manuscript presented in an intelligible fashion and written in standard English?

Reviewer #1: Yes

6. Review Comments to the Author

Reviewer #1: Thank you for making suggested changes. The manuscript is clear and now improved. All comments have been addressed.

7. PLOS authors have the option to publish the peer review history of their article (what does this mean?). If published, this will include your full peer review and any attached files.

Reviewer #1: No

---

## [Editor Report · Acceptance letter]

3 May 2021

PONE-D-20-37553R2 

Effect of belt electrode-skeletal muscle electrical stimulation on immobilization-induced muscle fibrosis 

Dear Dr. Okita:

I'm pleased to inform you that your manuscript has been deemed suitable for publication in PLOS ONE. Congratulations! Your manuscript is now with our production department. 

Kind regards, 

on behalf of

Dr. Aldrin V. Gomes 

Academic Editor

PLOS ONE